# Oxidation of Supported Nickel Nanoparticles at Low Exposure to O_2_: Charging Effects and Selective Surface Activity

**DOI:** 10.3390/nano12071038

**Published:** 2022-03-22

**Authors:** Andrey K. Gatin, Sergey Y. Sarvadii, Nadezhda V. Dokhlikova, Vasiliy A. Kharitonov, Sergey A. Ozerin, Boris R. Shub, Maxim V. Grishin

**Affiliations:** N. N. Semenov Federal Research Center for Chemical Physics, Russian Academy of Sciences (FRCCP RAS), Kosygina Street 4, 119991 Moscow, Russia; akgatin@yandex.ru (A.K.G.); dohlikovanv@gmail.com (N.V.D.); vch.ost@mail.ru (V.A.K.); sergeoz@yandex.ru (S.A.O.); bshub@mail.ru (B.R.S.); mvgrishin68@yandex.ru (M.V.G.)

**Keywords:** nanoparticles, nickel, highly oriented pyrolytic graphite, oxygen, adsorption, dissociation, scanning tunneling microscopy, scanning tunneling spectroscopy

## Abstract

The oxidation of Ni nanoparticles supported on highly oriented pyrolytic graphite was investigated under conditions of low exposure to oxygen by methods of scanning tunneling microscopy and spectroscopy. It was found that charge transfer effects at the Ni-C interface influenced the surface activity of the nanoparticles. The O_2_ dissociation and the Ni oxidation were shown to occur only at the top of the nanoparticle, while the border of the Ni-C interface was the less preferable area for these processes. The O_2_ dissociation was inhibited, and atomic oxygen diffusion was suppressed in the given nanosystem, due to the decrease in holes concentration.

## 1. Introduction

Today, nickel-based catalytic systems are of great interest as nickel is widely used as catalyst for selective hydrogenation of poly-unsaturated compounds in petrochemistry [1,2], steam reforming [3], carbon nanotubes synthesis [4], nitro compounds reduction [5,6] and other processes. They are one of the most intensively explored catalysts, and the mechanisms of the reactions on its surface are studied in detailed.

At the same time, reaction mechanisms are known to change significantly while the catalyst is scaled up to a nanosized system, where the size effects might take place and influence the dissociation of gas molecules, inhibiting or promoting reactions over the surface of metal nanoparticles [7]. Adsorption and dissociation are the first stages of any surface reaction, and they can change significantly in nanostructured systems, due to such factors as local non-stoichiometry of chemical composition of the surface, inhomogeneity of its atomic structure, charge redistribution, etc. For example, charge redistribution was shown to be the key factor in the hydrogenation of gold nanoparticles supported on highly oriented pyrolytic graphite (HOPG) [8]. Such result has demonstrated the ability to control the rate of hydrogenation and demonstrates a very good agreement with other works [9]. At the same time, the interaction of gold nanoparticles with the support and consequent charge redistribution result in the selective surface activity in the hydrogenation of gold nanoparticles [10]. Charge redistribution also significantly influences the orientation of polar gas molecules and, consequently, the reaction rate. The inhomogeneity of the surface atomic structure can enhance this effect. As was shown for the reduction of copper-based nanoparticles in CO, it was the curvature of nanoparticle surface that determined the electric field strength and the ratio of oriented CO molecules in the vicinity of the nanoparticle surface [11]. Additionally, even more, some catalytic effects were shown to be explained only by the charge redistribution influenced by the electrochemical environment [12]. Previously, we have already studied gas reactions over Ni-based nanostructured elements of bicomponent coatings [13], but recent experiments with gold nanoparticles [10] have also indicated the urgency to study the initial stages of the nickel nanoparticles oxidation.

So, we can say “the size matters”, and initial stages of surface reactions should be elucidated even for such well-studied catalysts as nickel-based ones while scaling them up to nanosized systems. The aim of our work was to find out the correlation between the activity of the O_2_ dissociation sites and the charge transfer in the vicinity of Ni-C interface. Since any catalyst is a dynamic system that responds to changes in external factors, the results of our research are of significant practical importance for fine-tuning the catalyst, taking into account its interaction with the smallest oxygen impurities in the reacting gas mixtures.

## 2. Materials and Experiment

The experiments were carried out in the vacuum chamber of a scanning tunneling microscope (UHV VT STM, Omicron NanoTechnology, Taunusstein, Germany) under ultrahigh vacuum conditions. So, the uncontrolled changes in the chemical composition of the samples due to residual gases were avoided. The UHV chamber was also equipped with gas-pumping system, the pipelines for gas-injection, and quadrupole mass spectrometer (HAL 301 PIC, Hiden Analytical Limited, Warrington, UK) for gas purity control. The residual pressure in UHV chamber did not exceed 10^−10^ mbar. The sample exposure was measured in Langmuir (1 L = 1.33 × 10^−6^ mbar·s).

The STM tips used for the experiments were made of tungsten wire by electro-chemical etching and treated by argon-ion sputtering under UHV conditions. Only those tips were used that demonstrated a reproducible S-shaped curve of the volt-ampere characteristic (VAC) when scanning HOPG. Such form of VAC curve is typical for metal–metal tunnel nanojunctions [14].

Nickel nanoparticles were synthesized by the impregnation–precipitation method on the surface of highly oriented pyrolytic graphite with an angular spread of the *c*-axes of the crystallites of 0.8° [15]. The surface of cleaned prepared HOPG plates (AIST-NT, Moscow, Russia) looked like vast atomically smooth C(0001) terraces. An aqueous solution of nickel(II) nitrate Ni(NO_3_)_2_ with nickel concentration of 25 mg/L was used as nanoparticles precursor. A drop of this solution was applied on the HOPG surface and dried. Then, the sample was calcined for 40 h in hydrogen (P = 10^−6^ mbar, T = 950 K) in the ultrahigh vacuum (UHV) chamber of the scanning tunneling microscope (STM). During the calcination, the nickel nitrate decomposed with the formation of solid nickel oxide and gaseous nitrogen oxide and oxygen, which were pumped out of the chamber. The residual gas pressure in the UHV chamber did not exceed 2 × 10^−10^ mbar.

The final stages of nickel precursor decomposition occur according to the scheme [16]:(1)$$\mathrm{Ni}{\left({\mathrm{NO}}_{3}\right)}_{2}\overset{\sim 573\;K}{\to }\mathrm{NiO}+O_{2}+{\mathrm{NO}}_{2},$$
while the reduction of nickel(II) oxide in hydrogen can be described as [16]:(2)$$\mathrm{NiO}+H_{2}\overset{\sim 535\;K}{\to }\mathrm{Ni}+H_{2}O.$$

As the calcination temperature exceeded both the decomposition temperature and the reduction temperature, one can conclude that calcination under such conditions is enough for complete oxide reduction and pure nickel formation.

After the calcination the residual H_2_ was pumped out of the vacuum chamber and the result of nanoparticles synthesis was determined by scanning tunneling microscopy and spectroscopy (STM/STS) methods. In order to carry out the oxidation experiment the sample was exposed to oxygen (50 L) at T = 300 K and then it was left in UHV chamber for 24 h till complete relaxation of the system is ended. Then, the STM/STS measurements were performed, and the result was compared with the same one for the initially synthesized system.

## 3. Results and Discussion

Results of the STM measurements show that after calcination nanoparticles are formed on the HOPG surface with an average lateral size of 5–6 nm. An example image of the HOPG surface area with nickel nanoparticles deposited on it is shown in Figure 1a. Nanoparticles are agglomerated in clusters and decorate the edges of HOPG terraces. The nanoparticles shape is close to an oblate half-spheroid, and their average height over the atomically smooth HOPG terrace is 0.8–3 nm (see Figure 1b).

Analysis of the spectroscopy data demonstrates that the surface of nanoparticles possesses metal electronic structure after calcination. One can see that VAC curves both for nanoparticles and HOPG are S-shaped and correspond to the one typical for metal–metal tunneling junction (see Figure 2).

In our previous experiments even the exposure of 200 L was found to be enough for complete oxidation of nickel nanoparticles [13]. So, this time we decided to reduce the exposure to 50 L (P = 10^−7^ mbar). The VAC curves of HOPG have stayed the same after the exposure to oxygen and 24 h of the relaxation of the synthesized system, but the electronic structure of the nanoparticles has changed significantly. According to the STS data the surface of nickel nanoparticles has become composite. Figure 3 demonstrates the areas of the nanoparticle surface where two different types of electronic structure can be observed. A region of zero current appears at VAC curves measured at the central part of the nanoparticle. Up to the dimensional factor the width of this region coincides with the band gap of the surface material [17]. So, we can conclude that after oxidation, the central part of the nanoparticle has changed its electronic structure from metallic to semiconducting, and the band gap of the synthesized oxide compound is 1.1–1.8 eV. At the same time, this semiconducting area is surrounded by the ring of pure metal spanning along the nanoparticle’s edge. Thus, we see that at low exposure to O_2_ the oxidation of nickel nanoparticles is affected significantly by the nickel–carbon interface.

We should consider if the changes in the electronic structure of the nanoparticles are related to their oxidation. Indeed, the 24 h of the system relaxation under UHV conditions at 300 K allows one to estimate the lower limit of desorption energy *E_d_*. According to the Frenkel formula [18]:(3)$$\tau =\tau_{0}\cdot exp\left(E_{d}/kT\right),$$
where *τ*_0_ is ~10^13^ s, *k* is Boltzmann constant, and *τ* is the duration of system relaxation at temperature *T*. Thus, one can estimate *E_d_* to be ≥ 0.5 eV. Unfortunately, this value does not provide any strict conclusion if the adsorption is chemical or physical. At the same time, the oxygen chemisorption on nickel is known to occur even at 77 K and pressure of 10^−8^ mbar [19,20]. So, there is no doubt that observed changes in nickel electronic structure are due to its chemical interaction with oxygen.

Measuring the width of zero-current region at VAC curves, one can establish that nickel band gap after interaction with O_2_ is 1.5 eV. This value is surprisingly low, while for the most stable oxide NiO the reported optical band gap ranges from 3.4 eV to 4.3 eV [21]. At the same time, this value coincides well with the band gap of NiO nanoparticles synthesized under the same conditions without nickel reduction [22]. Of course, we should remember that nickel tends to form a range of nonstoichiometric oxides and solid solutions containing both Ni^2+^ and Ni^3+^ ions, and this may result in changes in chemical and electronic features of Ni-based systems [23,24,25].

As mentioned above, the STS results demonstrate significant inhomogeneity in oxide distribution on the surface of the nanoparticle. The question arises about the reason for this fact. One should elucidate if the oxide distribution coincides with the location of adsorption active sites or dissociation active sites, taking in account the possibility of surface migration, volume diffusion and other factors.

We can define some reasons for significantly inhomogeneous oxide distribution: (i) initial chemical inhomogeneity of the nanoparticle; (ii) diversity in the initial atomic structure of the facets terminating the nanoparticle surface; (iii) charge redistribution within the nanosystem. Let us consider all these variants sequentially.

### 3.1. Initial Chemical Composition

As we have mentioned above, after calcination in hydrogen, the nanoparticles surface obtains an electronic structure of metal type. According to the STS results, there was no VAC curve with zero current region within the nanoparticles surface—it was completely metallized. So, one can conclude that inhomogeneity in oxide distribution cannot result from the initial deviations of the surface chemical composition.

Additionally, what is it about the inside composition of the nanoparticles? The STM/STS methods are sensitive to the surface, but they are not able to ‘look inside’ the nanoparticle. As they were synthesized from nickel oxide, we can suppose that inhomogeneous distribution of surface oxide is due to the residual oxygen inside them. So, the question arises if there is an oxide core inside the nanoparticle which can influence the surface reactions. Simple diffusion estimates show that for the hydrogen atom, the root-mean-square path *d* under given reduction conditions is five orders greater than the average nanoparticle size. Indeed, we know that:(4)$$D=D_{0}\cdot exp\left(-E_{a}/RT\right),$$
where *D*_0_ is ~10^−10^ sm^2^/s and *E_a_* is ~0.3 eV for hydrogen diffusion in metals [18]. So, one can count *D* is ~10^−12^ sm^2^/s under given calcination conditions and after 40 h we have:(5)$$d={\left(2Dt\right)}^{1/2}=\sim 3\times 10^{-3}\;\mathrm{sm}.$$

Therefore, the observed nickel reduction at the surface can guarantee complete nickel reduction inside the nanoparticles.

We can conclude that inhomogeneity in the surface distribution of nickel oxide cannot be explained either by the surface chemical composition or by the initial volume composition of the nanoparticles.

### 3.2. Atomic Structure

Selective chemical activity of the nanostructured surface can result from the difference in atomic structure of facets terminating the nanoparticles surface. Indeed, the mechanisms of oxygen adsorption can vary from facet to facet even for the crystal with the same lattice [26].

Of course, such an effect manifests itself significantly for small atomic clusters of regular shape. The bigger the nanoparticle, the less significant the role of this effect. One can estimate that the nanoparticle with a lateral size of 3–5 nm consists of ~3500 atoms. We can conclude that nanoparticles of such size are big enough, so the areas of atomically smooth plane surface may be the only source of selective surface activity, like in the case of monocrystals.

However, the profile shown at Figure 1b does not demonstrate any signs of such atomically smooth facets on the surface. Comparing the oxide distribution with the profile form, we can note that the oxide covers the surface with significant curvature. It means that inhomogeneous oxide distribution does not correlate with the atomic structure of the nanoparticle surface.

### 3.3. Charge Distribution

The last option is to consider the inhomogeneity of the charge distribution within the nanostructured system. The phenomenon of contact potential difference is known to manifests itself for dissimilar electrodes as a result of the difference in their work functions [27]. The work function of graphite is 4.7–4.8 eV [28], and that of nickel is about 5.04–5.35 eV [29]. We can conclude that the nickel nanoparticle will be charged negatively in the area of contact with graphite support. However, the area of excessive charge is limited to 2–3 layers of nickel atoms, as this effect damps quickly within conductive materials. It means that within the nickel nanoparticle, an area of increased local electron concentration appears in the vicinity of the Ni-C interface. In the case of the nanoparticles with half-spheroid shape, it will resemble the formation of a ring along the outside border of the nanoparticle, while the electronic structure will not change in the central part.

In the case of the nanoparticles with a more sophisticated shape, the oxide formation will occur depending on the number of the nickel atomic layers beneath the dissociation active site. If the nanoparticle is surrounded by an HOPG area which is covered with a thin nickel layer, the oxide formation will take place only at the top of the nanoparticle. The morphology of such mantle-possessing nanoparticles will be perceptible for STM measurements, but the mantle will not differ from HOPG in spectroscopy as it is free of any oxides. Subsystems with such sophisticated oxide distribution were found during the experiments. A good example of mantle-possessing nanoparticle is shown in Figure 4. This fact proves that the activity of the dissociation sites is affected indeed by the phenomena of contact potential difference.

At the same time, the oxygen adsorption on nickel is known to be dissociative with formation of various charged O^δ−^ species [30]. It is accompanied by electron transfer from metal into molecular oxygen antibonding *π**-orbitals and is completed by filling the antibonding *σ**-orbitals [18,31]. So, we should expect the result of dissociation to be reversed in relation to what is observed: excessive charge should promote the oxygen dissociation and oxide formation along the outside border of the nanoparticle and not to inhibit it. We watched for the same result in H_2_ dissociative chemisorption on the surface of HOPG-supported gold nanoparticles [11,32]. This contradiction demonstrates that the detailed mechanism of oxygen dissociation on nickel nanoparticles is actually still unclear in this case. It is very likely that not only should the enhanced electron concentration be considered, but also the mobility of the electrons, as it may decrease due to the presence of the positively charged carbon layer and polarization of the C-Ni interface.

In addition, the ring-centric pattern of the surface oxide distribution after exposure to oxygen—as we have noticed—is very likely to result from the volume pattern of excessive charge spread. Thus, we may conclude that the contact potential difference and the resulting charge redistribution significantly affect the oxygen dissociation. Even though we cannot point out which factor is the prevailing one—the increase in the local electron concentration due to the difference in work functions or the decrease in the electron mobility due to the presence of positively charged carbon layer—we can say that the observed pattern of oxide distribution actually images the distribution of oxygen dissociation sites over the nanoparticle surface.

However, the question arises if this image is not distorted by the surface migration of chemisorbed oxygen and its diffusion inside the nanoparticle. How do these processes influence the initial distribution of the dissociation active sites over the surface? The processes mentioned can be directional and ordered in the presence of charge transfer and polarization effects.

The surface migration takes place at room temperature and its activation energy is known to range from *E_d_*/5 to *E_d_*/3, where *E_d_* is desorption energy [33]. At the same time, the oxygen dissociative adsorption on the nickel surface is known to be accompanied by a significant rearrangement of the metal surface. The oxygen atom immediately swaps places with the lower nickel atom and penetrates into the metal lattice. Thus, the surface migration of chemisorbed oxygen does not occur in the given Ni-C nanostructured system and does not distort the observed distribution of dissociation active sites. All oxygen immediately ‘crawls’ under the upper layer of nickel atoms, and the size of the observed oxide spot does not change.

However, it is the question of diffusion in volume that remains. The STM/STS methods are sensitive to changes in morphology and electronic structure within several atomic layers at the surface of the sample. The diffusion of oxygen atoms beneath the nickel surface can affect the STM spectra. However, the situation is actually a bit more complicated, due to the fact that it is practically impossible to distinguish between the diffusion and the oxidation reaction. The formation of a wide range of nonstoichiometric oxides is the problem in this case.

The elements of the iron group are known to form mixtures of divalent and trivalent oxides and can be considered as solid solution M_x_O, where *x* varies from 0.97 to 1.70 in the case of nickel [34,35]. All of them are *p*-type semiconductors, so the diffusion of oxygen atoms through the oxide layer can be affected by the redistribution of charge carriers [23]. The increase in the electron holes concentration or saturation of the system with electrons will inhibit or promote the expansion of the oxide layer and influence the characteristic size of observed oxide spot.

The oxygen dissolution in nickel oxide layer is accompanied by the formation of cation vacancy *V_Ni_*—singly ionized and negatively charged—and positive hole *h* according to the equation [23]:(6)$$\frac{1}{2}O_{2}=O_{\mathrm{ox}}+V_{Ni}^{-}+h^{+},$$
where O_ox_ is oxygen dissolved within the oxide. The holes concentration depends on the oxygen pressure, as in [23]:(7)$$\left[h\right]=[V_{Ni}]\propto p_{O_{2}}^{\frac{1}{8}}.$$

Of course, the fulfillment of this ratio at room temperature is debatable, but in any case, we can conclude that the formation of new holes will slow down significantly after finishing the sample exposure and pumping the gas out of the vacuum chamber.

At the same time, the oxygen diffusion in nickel oxide is known to depend on the holes concentration: it is inhibited while there is a holes deficiency [23]. Addition of electron-donating elements into nickel oxide leads to the saturation of the system with electrons—an increase in the Fermi level [23]. We can suppose that the increase in the electrons concentration in the formed oxide can inhibit the diffusion of oxygen. This idea is consistent with data on oxygen diffusion in doped cobaltous oxide [36], in which it was demonstrated that the diffusion coefficient decreased in the order: lithium doped oxide, undoped, aluminum doped—corresponding to the relative order of concentration of anion vacancies. One can suppose that interaction of negatively charged species with anion vacancies may decrease the efficiency of oxygen diffusion via anion vacancies. However, the details of this mechanism should be elucidated.

As mentioned above, HOPG-supported nickel nanoparticles possess an increased local electron concentration in the vicinity of the contact area due to the contact potential difference. However, it is the saturation of the system with electrons that inhibits oxygen diffusion. That means that there are two factors inhibiting the oxygen diffusion in volume. The first one is the decrease in the oxygen pressure and pumping the gas out of the vacuum system that eliminates the source of oxygen molecules. The second one is the electron transfer and charging of nanoparticles with a negative charge. Both of these factors prevent the oxygen diffusion within the nanoparticle.

That is, the oxide distribution over the nanoparticle surface actually images the distribution of oxygen dissociation sites. The observed structure is not distorted by the surface migration of adsorbed O^δ−^ species or by the diffusion of oxygen atoms inside the nanoparticle.

## 4. Conclusions

The above experiments demonstrated that active sites of oxygen dissociation are located in the area which is distant from the Ni-C interface. The reason for such selective surface activity is the charge transfer in the vicinity of the contact area, due to the difference in electron work functions for HOPG and nickel. As a result, the dissociation occurs only beyond 2–3 atomic layers far from the Ni-C interface, where the charging effect is damping. The observed ring-centric pattern of oxide distribution contradicts the known mechanism of oxygen dissociation accompanied by the electron transfer from metal into molecular antibonding orbitals. So, the observed pattern unequivocally indicates the difference in electron work functions as the key factor, though the detailed mechanism of oxygen dissociation is still unclear.

The oxygen diffusion through the nickel oxide depends on the holes concentration. The higher the saturation of the system with electrons, the lower the holes concentration, and consequently, the worse the oxygen diffusion. So, the increased negative charge of the nickel nanoparticle prevents its complete and uniform oxidation and preserves the ring-centric pattern of oxide distribution. As the O_2_ dissociation is accompanied by simultaneous penetration of oxygen atoms into the nickel lattice, the surface migration of chemisorbed O species also does not occur.

Therefore, the oxide distribution images the actual location of oxygen dissociation active sites, and it is not distorted by the surface migration of chemisorbed atoms and O diffusion in the nanoparticle.

## Figures and Tables

**Figure 1 nanomaterials-12-01038-f001:**
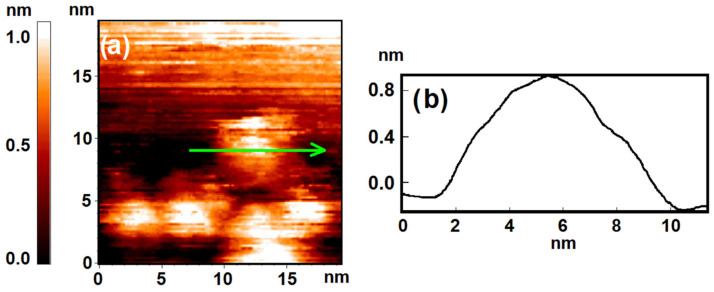
The sample after calcination under UHV conditions. Results of the STM measurement: (**a**) topography image of HOPG surface with deposited clusters of Ni nanoparticles; (**b**) profile of the surface along the cut line shown in (**a**).

**Figure 2 nanomaterials-12-01038-f002:**
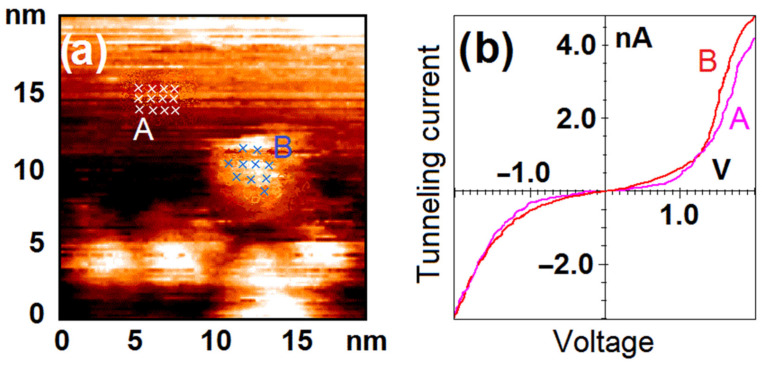
The sample after calcination under UHV conditions. Results of the STM/STS measurement: (**a**) topography image of HOPG surface with deposited clusters of Ni nanoparticles; (**b**) VAC curves of the tunneling currents averaged over the set of points on the surface of nanoparticles (red curve B) and HOPG (pink curve A) marked with crosses in (**a**).

**Figure 3 nanomaterials-12-01038-f003:**
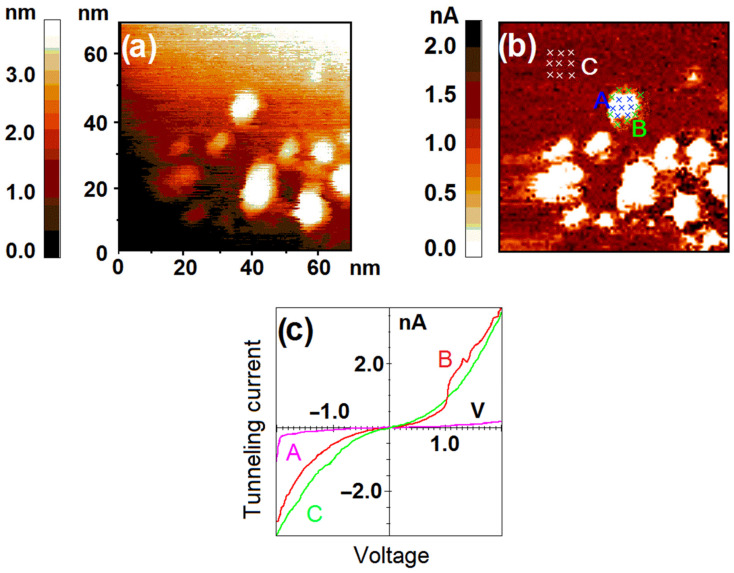
The sample after exposure to O_2_ (50 L). Results of the STM/STS measurement: (**a**) topography image of the HOPG surface with deposited nickel nanoparticles; (**b**) distribution of tunneling current values at 1.2 V over the same area with points of spectroscopy measurements marked with crosses; (**c**) VAC curves of the tunneling currents averaged over the set of points on the surface of HOPG (green curve C), central (pink curve A) and perimeter (red curve B) areas of the nanoparticle marked in (**b**).

**Figure 4 nanomaterials-12-01038-f004:**
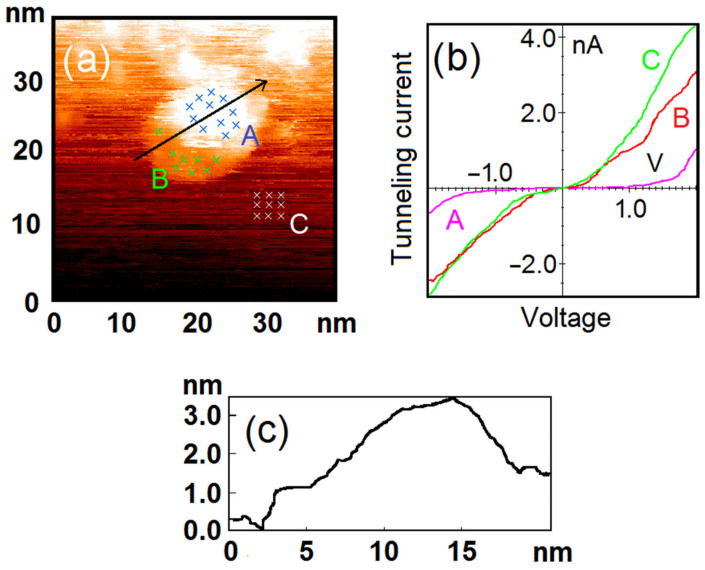
The mantle-possessing nanoparticle after exposure to O_2_ (50 L). Results of the STM/STS measurement: (**a**) topography image of HOPG surface with deposited nickel nanoparticle; (**b**) VAC curves of the tunneling currents averaged over the set of points on the surface of HOPG (green curve C), central (pink curve A) and mantle (red curve B) areas of the nanoparticle marked with crosses in (**a**); (**c**) profile of the nanoparticle surface along the cut line shown in (**a**).

## Data Availability

The data presented in this study are available on request from the corresponding author.

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
