# Peer review of "Oxidation of Supported Nickel Nanoparticles at Low Exposure to O2: Charging Effects and Selective Surface Activity"

_nanomaterials, 2022, doi:10.3390/nano12071038_

Round 1

Reviewer 1 Report

Attached file.

Author Response

Many thanks for your valuable comments that helped us to improve the quality of the manuscript! Our explanation and answer is attached.

Reviewer 2 Report

The authors investigated the correlation between the O2 dissociation sites and the charge transfer in the vicinity of the Ni-C interface of highly oriented pyrolytic graphite supported nickel nanoparticles. The nickel oxidation at low oxygen exposure was found be affect by the Ni-C interface and different reasons for inhomogeneous nickel oxidation have been considered. The manuscript is well written, and a careful explanation of the data reported is given.

Only minor corrections are required:

  • Second line, introduction section: “...nickel is widely used as catalysts” should be corrected as “nickel is widely used as catalyst”
  • Page 4: the sentence “In our previous experiments...” require a citation of appropriate reference
  • Page 9: in the sentence “Since our experiments were carried out at very low pressure” can the author give more information about the pressure used in their experiments?

Author Response

(The authors gave the same response as above.)

Reviewer 3 Report

This paper studies the oxidation of Ni nanoparticles supported on highly oriented pyrolytic graphite under conditions of low exposure to oxygen and reveals the correlation between the activity of the O2 dissociation sites and the charge transfer in the vicinity of Ni-C interface. This work provides detailed information and reliable conclusion on this subject. Thus, I suggest its publication in Nanomaterials after minor revision.
The followings are suggestions for improvement:
1. The introduction should provide sufficient background, please introduce more relevant research  progress and further elaborate the practical significance of the research.
2. HOPG appears in lines 38 and line 66, but you don't give an explanation until line 69, please recheck and provide some instructions about the  HOPG.

Author Response

(The authors gave the same response as above.)

Reviewer 4 Report

I have only one minor suggestion. Please at the first use of the dose of oxygen 50 L develop what means L.

Author Response

(The authors gave the same response as above.)
